# Sperm Ion Transporters and Channels in Human Asthenozoospermia: Genetic Etiology, Lessons from Animal Models, and Clinical Perspectives

**DOI:** 10.3390/ijms23073926

**Published:** 2022-04-01

**Authors:** Emma Cavarocchi, Marjorie Whitfield, Fabrice Saez, Aminata Touré

**Affiliations:** 1Institute for Advanced Biosciences, INSERM U1209, CNRS UMR5309, Université Grenoble Alpes, 38000 Grenoble, France; emma.cavarocchi@inserm.fr (E.C.); marjorie.whitfield@inserm.fr (M.W.); 2UMR GReD Institute (Génétique Reproduction & Développement) CNRS 6293, INSERM U1103, Team «Mécanismes de L’Infertilité Mâle Post-Testiculaire», Université Clermont Auvergne, 63000 Clermont-Ferrand, France

**Keywords:** sperm, asthenozoospermia, gene mutation, ion channel, mouse, human

## Abstract

In mammals, sperm fertilization potential relies on efficient progression within the female genital tract to reach and fertilize the oocyte. This fundamental property is supported by the flagellum, an evolutionarily conserved organelle that provides the mechanical force for sperm propulsion and motility. Importantly several functional maturation events that occur during the journey of the sperm cells through the genital tracts are necessary for the activation of flagellar beating and the acquisition of fertilization potential. Ion transporters and channels located at the surface of the sperm cells have been demonstrated to be involved in these processes, in particular, through the activation of downstream signaling pathways and the promotion of novel biochemical and electrophysiological properties in the sperm cells. We performed a systematic literature review to describe the currently known genetic alterations in humans that affect sperm ion transporters and channels and result in asthenozoospermia, a pathophysiological condition defined by reduced or absent sperm motility and observed in nearly 80% of infertile men. We also present the physiological relevance and functional mechanisms of additional ion channels identified in the mouse. Finally, considering the state-of-the art, we discuss future perspectives in terms of therapeutics of asthenozoospermia and male contraception.

## 1. Introduction

Spermatozoa constitute one of the most highly differentiated cell types in sexually reproducing organisms. Indeed, sperm cells show a highly compacted nucleus to protect their genetic material, sophisticated cytoskeletal machinery to enable flagellar beating, and complex signaling and metabolic pathways, which sustain their survival and progression throughout their journey in the male and female genital tracts and, overall, promote their fertilization potential. Among the regulatory pathways and functional mechanisms involved in their maturation and fertilization potential, ion fluxes and homeostasis play a remarkable role. This is illustrated, for example, by flexible regulation of the cytoplasmic volume of the sperm cells in response to hypo-osmotic stress encountered during their transit from the testicular efferent ducts to the epididymis [1,2]. Another important role of ion modulation in sperm function is observed during their post-testicular maturation, which occurs within the epididymis and the female genital tract, ultimately conferring sperm motility and fertilization potential (for review, see [3]). As a consequence, the dysregulation of ion fluxes in sperm cells has long been demonstrated, both in humans and animal models, to impair the regulation of sperm volume and overall sperm progression and fertilization capacity. Importantly, in humans, asthenozoospermia, defined by the reduction or absence of motile sperm in the ejaculate, constitutes the predominant sperm pathological condition and is observed in nearly 80% of infertile men (for review, see [4]).

To date, although the clinical and genetic diagnosis of asthenozoospermia has greatly improved (for reviews, see [5,6]), the only possible treatment provided to infertile men consists of assisted reproduction technologies (ART), in particular, intracytoplasmic sperm injection (ICSI). The identification of potent strategies to stimulate sperm motility and fertilization potential could therefore constitute an alternative to ICSI by the direct treatment of infertile men. On the other hand, targeting proteins specifically localized at the surface of sperm cells could also be a strategy for innovative male contraception. In this context, this systematic literature review is focused on describing the plasma membrane proteins that act as ion transporters or channels involved in the regulation of sperm motility and fertilization potential. We will first provide an update on such proteins for which mutations have been formally demonstrated in humans to be a cause of male infertility due to asthenozoospermia and which are therefore essential for sperm motility and fertilization potential. We will next highlight other ion channels and transporters for which the physiological relevance and molecular mechanisms of action have been established in mouse models and which could consequently also be important in humans. Finally, we will describe the state-of-the-art on the identification of specific ion transmembrane channels delivered by epididymosomes during sperm epididymal transit that constitute attractive targets for the treatment of asthenozoospermia and the development of male contraceptives.

## 2. Ion Exchange and Signaling Pathways in Sperm Motility and Capacitation

Spermatozoa released from the testis are morphologically differentiated but remain immotile and are unable to recognize and fertilize the oocyte unless they undergo a series of maturation events that occur during their transit through the male and female genital tracts [7,8]. The primary motility of spermatozoa is acquired during their transit through the intermediate segment of the epididymis before storage in a quiescent state in the caudal region of the epididymis. After ejaculation and upon exposure to the microenvironment of the female genital tract, spermatozoa are activated by a process called capacitation, which increases the amplitude and velocity of flagellar beating (i.e., hyperactivation) and ultimately enables sperm penetration through the egg zona pellucida and gamete recognition (for review, see [9]). 

It is well established that post-testicular functional maturation of sperm cells relies on the micro-environment of the genital tracts and is tightly controlled by nutrient availability, pH, and overall ion content (for review, see [3]). In particular, within the epididymis, the establishment of a low bicarbonate (HCO_3_^−^) concentration contributes to luminal acidification, which is necessary for sperm maturation and subsequent storage in a quiescent state (for review, see [10]). Following ejaculation, sperm cells are exposed to the basic pH of the female genital tract and undergo biochemical and electrophysiological changes (for review, see [11]) that are indispensable for the hyperactivation of sperm motility and acquisition of fertilization potential; this process, discovered by Chang and Austin in the early 50s, is called capacitation (for reviews see [11,12]). 

Subsequently, a panoply of ion transporters located at the surface of the sperm cells have been characterized and shown to be involved in a complex series of ion exchanges that regulate sperm fertilization potential (for reviews, see [13,14,15,16]). Their involvement and sometimes absolute requirement for the acquisition of primary motility during epididymal transit and capacitation have been supported by numerous studies of mutant mouse models. A number of the signaling events that promote primary motility during sperm epididymal transit have also been deciphered (i.e., protein phosphorylation cascades involving WINT, the GSK3 kinase, and PPP1 and PPP2 phosphatase activation) [3,17]. However, the molecular mechanisms that possibly connect transmembrane ion fluxes to these signaling pathways are yet to be defined. By contrast, the molecular mechanisms that occur during sperm capacitation in the female genital tract have been better documented. Hence, in response to genital ion content, functional cooperation between ion channels and transporters has been shown to induce the increase in plasma membrane fluidity, alkalinization of the cytoplasm, and activation of protein phosphorylation cascades that are essential for sperm hyperactivation and fertilization potential (for reviews see [9,11]) (Figure 1). Among these phosphorylation events, the protein kinase A (PKA) pathway has been exhaustively documented and constitutes a hallmark of sperm capacitation in many species. Hence, Ca^2+^ and HCO_3_^−^ were shown to directly bind to the soluble adenylate cyclase (ADCY) and stimulate cAMP production [18,19]. The resulting increase in intracellular cAMP concentration allows activation of PKA and the subsequent phosphorylation cascades of flagellar proteins that are indispensable for sperm fertilization (for review, see [20]). It was proposed that fast and slow activation events of the PKA pathway are involved in the activation of primary motility and capacitation associated with tyrosine phosphorylation, respectively, through the recruitment of various HCO_3_^−^ channels and transporters (for review, see [21]). The calcium/calmodulin-dependent protein kinases (CaMKs) have also been shown to be involved in sperm capacitation through their interaction with calmodulin (CaM), a well-established calcium-binding and regulatory protein present in the sperm head and flagellum (for review, see [22]). Among the targets of these phosphorylation cascades, both axonemal and peri-axonemal proteins of the sperm flagellum have been identified, such as AKAP scaffold proteins [23], enzymes involved in energy metabolism (pyruvate dehydrogenase, aldolase) [24], that are localized at the fibrous sheath of the sperm flagellum, and axonemal proteins, which orchestrate flagellar beating (tubulin [24] and dynein chains [25]); the exact consequences and translation of these phosphorylation events into specific functional properties have not yet been elucidated. Overall, although most of the associated molecular mechanisms are yet to be uncovered, the correct adaptative response of sperm cells to the ion content and pH of both the male and female genital tracts appears to be absolutely essential for the regulation of flagellar beating and ultimately, the promotion of sperm progression and hyperactivation during capacitation.

## 3. Mutations in Ion Transporters and Channels That Result in Human Asthenozoospermia

In recent years, it has been estimated that approximately 48.5 million couples worldwide are confronted with pregnancy failure [27], and this social and medical issue is becoming increasingly urgent. A male factor is involved in nearly half of cases of couple’s infertility, corresponding to 7 to 10% of men of the world population suffering from infertility [6,28]. As male infertility is a complex multifactorial pathology, it often remains unexplained; in particular, although genetic factors are inferred as potential frequent causes (at least 15% of cases; [29]), to date, only a minority of cases can be assigned to a precise genetic defect [28].

Among the causes of male infertility, asthenozoospermia is the predominant condition and is found in more than 80% of infertile men [30]. This pathological condition corresponds to the reduction or absence of motile spermatozoa in male ejaculate, with a lower reference value established by the World Health Organization of 40% of total motile sperm and 32% of progressive sperm. Asthenozoospermia is most often observed in association with quantitative and/or morphological sperm defects, called oligo-asthenozoospermia and astheno-teratozoospermia, respectively. To date, the majority of identified genetic causes correspond to asthenoteratozoospermia resulting from moderate or severe morphological defects of the sperm flagellum. In particular, a tremendous body of work has been carried out during the last decade on the genetic investigation of multiple morphological abnormalities of the sperm flagellum (MMAF), which correspond to the presence of a mosaic of sperm with short, angulated, or absent flagella or flagella of irregular caliber (for reviews see [31,32,33]). These achievements are of significant importance for the care provided to patients but they cannot be easily translated to potential therapeutic strategies aiming to rescue defects of flagellar biogenesis. On the contrary, asthenozoospermia, due to functional defects in signaling pathways, provide a unique context that could be favorable for drug targeting and the direct treatment of infertility. In this context, a better definition of this pathological condition by the investigation of the causal genetic defects and associated pathophysiological mechanisms is essential and will drive future clinical development in this field. In this chapter, we focus, in particular, on transmembrane ion channels and transporters located on the surface of sperm cells, for which gene mutations or genomic defects have been reported to cause asthenozoospermia. These channels are described below and summarized in Table 1.

### 3.1. CatSper

CATSPER (cation channel sperm associated) is a multi-protein channel specifically expressed in the testis and located at the plasma membrane of the principal piece in human and mouse sperm flagella, where it has been shown to be required for sperm hyperactivation. It is composed of a hetero-tetrameric pore (*CATSPER1–4*) associated with five auxiliary subunits (β, γ, δ, ε, ξ) that participate in stabilizing the complex, together with the pH-dependent Ca^2+^ sensor EFCAB9 [50,51]. Additional components have been discovered over the last several years, among them, CATSPER η and the testis-specific organic anion transporter SLCO6C1, which led to the term “CatSpermasome” to name the channel-transporter super-complex that mediates Ca^2+^ fluxes in sperm [52]. The CATSPER channel has been shown to mediate Ca^2+^ during sperm motility and capacitation in a cAMP- and pH-dependent manner [53]. In addition, in humans, the CATSPER channel has been shown to be activated by progesterone through direct binding to the channel [54,55], resulting in intracellular (Ca^2+^) oscillations [56]. Interestingly, recent work has highlighted the peculiar subcellular localization of CATSPER, which spans the plasma membrane of the flagellar principal piece with four nanodomains associated two by two in longitudinal rows [57].

CATSPER is the first ion channel for which gene defects were demonstrated to affect sperm motility in humans and result in male sterility. A partial genomic deletion of *CATSPER2* was initially described in a French family presenting with a deafness-infertility syndrome (DIS) with asthenozoospermia [35,36], followed by the identification of full deletions of the gene in three unrelated Iranian families [37,38]. Intriguingly, Luo et al. reported a copy number variation leading to heterozygous *CATSPER2* deletion in a Chinese patient showing a normal spermogram but with impaired sperm hyperactivation and zona pellucida penetration [39]. Avenarius et al. [34] also reported two consanguineous Iranian families segregating autosomal-recessive non-syndromic male infertility, characterized by asthenozoospermia, low sperm count, and abnormal sperm morphology. Analysis of the *CATSPER1* sequence in affected family members identified two insertion mutations, c.539–540insT, and c.948–949insATGGC, which led to frameshifts and premature stop codons. Finally, the importance of the CATSPER auxiliary subunits was supported by the study of a patient affected by idiopathic infertility and loss of CATSPER function for whom a deleterious in-frame deletion in *CATSPERε* was detected [40]. 

Consistent with the above findings in humans, *CatSper1* and *CatSper2* null mice were each shown to lack the other subunit, suggesting the importance of subunit co-expression [58]. An important functional consequence of hyperactivated motility is to promote sperm detachment from the epithelial reservoir sites of the oviduct for them to progress towards the oocyte. In this regard, Ho et al. showed that *CatSper1*/*2*-null sperm cells fail to detach from the female genital tract epithelium and are gradually eliminated from the oviduct [59]. Importantly, in the mouse, the invalidation of any of the four pore-forming α subunits gene, *CatSper1–4*, or the auxiliary *CatSper**δ* subunit gene was shown to cause asthenozoospermia by abrogating Ca^2+^ influxes and hyperactivated motility [53,60,61,62]. This suggests that, in humans, mutations in *CATSPER* genes other than those so far identified (i.e., *CATSPER1*, *2*, and *ε*) could be responsible for asthenozoospermia.

### 3.2. SLC26A3 and SLC26A8

SLC26 family members are transporters of small anions, including Cl^−^, HCO_3_^−^, sulfate, oxalate, and formate, and display wide tissue distribution. Most SLC26 isoforms act as coupled anion transporters, but some can mediate uncoupled electrogenic transport resembling that of Cl^–^ channels (see review [63]). SLC26 members share a common structure, with a transmembrane core that allows ion fluxes and a cytoplasmic region involved in protein trafficking and signaling. In particular, the STAS (sulfate transporter and anti-sigma factor antagonist) domain is evolutionarily conserved and present in bacterial orthologues; in some cases, a PDZ domain conferring scaffold properties is also present. Physical interaction of SLC26 members with the cystic fibrosis transmembrane conductance regulator (CFTR) channel has been highlighted by several studies. This interaction involves the SLC26 STAS domain and the R domain of the CFTR channel, mainly resulting in the stimulation of CFTR activity (for reviews, see [64,65,66,67]).

In humans, *SLC26s* are known to be important for the physiology of various organs, and gene mutations in *SLC26s* have been shown to be associated with several autosomal recessive disorders, including diastrophic dysplasia (*SLC26A2*), congenital chloride diarrhea (*SLC26A3*), Pendred syndrome (*SLC26A4*), and deafness (*SLC26A5*) (for reviews, see [68,69]). Importantly, knockout (KO) and knock-in mouse models have been generated for most SLC26 members and reproduce the clinical features of SLC26 human-related diseases when applicable (for review, see [67]). Relevant to this review, the SL26A3, A6, and A8 proteins are expressed in male reproductive organs and sperm cells. Importantly pathogenic gene variants in *SLC26A8* and *SLC26A3* were reported to induce human asthenozoospermia and male subfertility.

*SLC26A8: SLC26A8* (also called *testis anion transporter 1*, *TAT1)* shows remarkable tissue-specific distribution restricted to human and mouse testis [70,71]. The *SLC26A8* protein is exclusively detected in the male germline and localizes to the annulus and the equatorial segment of spermatozoa [71,72]. Heterozygous *SLC26A8* variants were shown to be associated with human male infertility in two patients with moderate asthenoteratozoospermia and a third characterized by severe asthenoteratozoospermia and reduced sperm count [43]. In vitro, these three missense variants (p.Arg87Gln, p.Glu812Lys, and p.Arg954Cys) appeared to impair protein stability and the mutated proteins failed to stimulate CFTR activity due to proteasomal degradation of the SLC26/CFTR protein complex [43]. These results are consistent with the role of *SLC26A8* in sperm morphology and functionality that were shown after total gene invalidation in the mouse model. Hence *Slc26a8* null mice showed infertility due to the complete loss of sperm motility and impaired fertilization potential—capacitation and acrosome reaction [71]. As *SLC26A8* localizes to the sperm annulus, the sperm morphology of the KO mice was mainly altered at the level of this structure, which appeared to be mislocalized and incomplete. Given its importance in delimiting sperm flagellar compartments and establishing a physical barrier along the flagellum, the defective annulus was associated with midpiece disorganization, an abnormally shaped mitochondrial sheath, and mostly flagellar angulations and hairpins, which increased in frequency as the sperm progressed through the epididymal tract. 

*SLC26A3*: Mutations in *SLC26A3* have been shown to be associated with congenital chloride diarrhea (CLD), a genetic disorder consisting of hypochloremia, alkalosis, and diarrhea, with V317del being the most common variant in the Finnish population [73]. A first study reported three CLD patients carrying this mutation and showing male subfertility [41]. In particular, the phenotype consisted of oligoasthenoteratozoospermia, low pH, a high Cl^−^ concentration in the seminal plasma, and spermatic cysts but normal spermatogenesis [41]. More recently, Wedenoja et al. reported two heterozygous variants in *SLC26A3* (p.Asp688His, p.Val318del) in a cohort of Finnish patients affected by CLD-associated infertility [42]. Interestingly, the p.Asp688His mutation was shown to affect the STAS domain and alter SLC26A3-CFTR complex activity without impairing ion exchange [42]. 

Consistent with the human phenotype, *Slc26a3* KO mice display gastrointestinal issues, such as diarrhea, altered tissue homeostasis in the colonic mucosa, and growth retardation [74]. Importantly, El Khouri et al. analyzed the status and functionality of the male reproductive system in *Slc26a3* null mice and observed both epididymal and sperm defects, whereas spermatogenesis in the testis appeared to be normal [75]. More precisely, the caudal region of the epididymis was the most affected in size and cytoarchitecture, with frequent epithelial vacuoles and granulomas. In the caput region, increased staining of V-ATPase proteins, which are markers of narrow/clear cells, was observed, consistent with their known modulation by bicarbonate and pH. Spermatozoa of *Slc26a3* null mice were less abundant and morphologically impaired (i.e., coiled or hairpin-shaped flagella, various head abnormalities). In addition, *Slc26a3* null sperm showed no progressive motility and did not respond to capacitation stimuli. Overall, this study highlighted the role of the *SLC26A3* anion transporter within the epididymal epithelium to regulate fluid composition and promote sperm maturation and within the sperm cells themselves to mediate sperm motility and capacitation processes [75]. The phenotype of oligo-asthenoteratozoospermia observed in *Slc26a3* null mice is in complete accordance with the reproductive phenotype observed in patients suffering from CLD.

Of note, most of the above variants identified in *SLC26A8* and *SLC26A3* were found at the heterozygous state, in contrast to mutations identified at the homozygous state in other SLC26 members, which follow an autosomal recessive mode of inheritance. This difference could be related to the haploid state of the germ cells; indeed, although permanent connection through intercellular bridges allows sharing between the spermatogenic cells, it could be uncompleted and could contribute to a heterogeneous sperm cell population in a given individual. Consistent with this, most individuals carrying heterozygous variants in *SLC26A8* and *SLC26A3* also presented with mild asthenozoospermia phenotype. In addition, as the heterozygous variants identified in *SLC26A8* and *SLC26A3* are missenses, they could also behave as a dominant negative effect and, therefore, be pathogenic at the heterozygous state.

Lastly, although SLC26A6 protein was reported to be present with *SLC26A3* and *A8* in sperm cells [76], its function in the mouse was shown to be dispensable for spermatogenesis and sperm fertilization potential [75] and, to date, no gene mutations in *SLC26A6* have been shown to be associated with human male infertility. 

### 3.3. SLC9C1 (sNHE)

Very recently, the importance of *SLC9C1* in human functional asthenozoospermia was confirmed by the finding of a homozygous truncating mutation in an infertile patient [44]. Before this discovery, the importance of *SLC9C1* in sperm function had only been explored in model organisms. Wang et al. first described this protein in mouse spermatozoa, where it is abundantly expressed, together with two other Na^+^/H^+^ exchangers (NHE1 and NHE5) [77]. *SLC9C1* belongs to the SLC9 solute carrier family and shows sperm-specific expression, which led to its second name: *sNHE* (sperm Na^+^/H^+^ exchanger). The main peculiarity of *SLC9C1* (*sNHE*) resides in its cytoplasmic tail, attached to the canonical channel pore, which is comprised of a cyclic-nucleotide binding domain (CNBD) and a voltage-sensing domain (VSD) involved in regulation. *SLC9C1* (*sNHE*) disruption in the mouse model caused infertility due to asthenozoospermia and capacitation defects, both phenotypes being partially rescued by sperm alkalinization and largely rescued by supplementation with cAMP analogs [77]. Consistent with these results, *SLC9C1* (*sNHE*) and soluble adenylate cyclase (sAC) have been reported to show co-dependent expression and to be mutually modulated by cAMP and pH/bicarbonate [78]. Interestingly the c.2748 + 2T > C splicing mutation identified in humans induces an in-frame deletion within the CNBD, which is likely to impair protein functionality [44]. The sperm of the patient carrying the *SLC9C1* mutation showed reduced total and progressive motility, as well as a number of morphological defects, such as disorganization of the midpiece and an increased fraction of isolated heads and break points after sperm selection, suggesting global fragility of the sperm structure [44].

*SLC9C1* (*sNHE*) has also been described in the sea urchin, which allowed Windler et al. to propose a role of an evolutionarily conserved chimera that combines both solute carriers and ion channels to ensure rapid regulation of the pH [79]. As cytoplasmic alkalinization activates the sperm calcium channel, it has been proposed that *SLC9C1* (*sNHE*), sAC, and CATSPER constitute a fundamental functional trio that regulates sperm motility and capacitation in Metazoans [80]. 

### 3.4. PKD (TRPP) 

*PKD* (polycystic kidney disease) genes, also known as *TRPP* (transient receptor potential polycystic), are mainly expressed in renal epithelial cells, and the encoded proteins are also detected in primary cilia and sperm flagella. The members of this protein family, namely *PKD1* and *PKD2*, form a putative heteromeric calcium channel. *PKD1* also harbors a G protein-coupled receptor autoproteolysis-inducing (GAIN) domain that may be responsible for its activation [81,82]. The role of the *PKD* gene in male fertility was investigated by a correlational study on patients suffering from autosomal dominant polycystic kidney disease (ADPKD) [49]. The authors analyzed a set of 90 unrelated families with ADPKD and identified predicted pathogenic variants in *PKD1* and *PKD2* associated with low sperm quality, mainly asthenozoospermia or oligo-astheno-zoospermia [49]. Although no further exploration of the impact of the identified *PKD* variants on protein functionality was performed [49], additional findings supporting an essential role of *PKD* genes for sperm motility were obtained in the Drosophila model. Hence, silencing of the *PKD2* ortholog, of which the expression is enriched in male reproductive organs, was shown to induce male infertility due to the inability of the spermatozoa to correctly find their way through the female genital tract and reach the sperm storage organs of the female flies [83,84,85]. The above data suggest a potential role of *PKD2* in sensing the environment of cilia and flagella. However, a recent study also reported the localization of TRPP2 in porcine sperm and identified functions in regulating intracellular Ca^2+^ and motility [86]. Blocking TRPP2 by incubating the sperm with TRPP2 antibodies also prevented Ca^2+^ influxes during capacitation, consistent with TRPP2 functioning as a plasma membrane cation channel [86].

### 3.5. VDAC2 and VDAC3

VDACs (voltage-dependent anion-selective channels) are known to mediate anion fluxes in the open state and cations in the closed state [87]. VDACs also behave as Ca^2+^-binding proteins and can mediate transmembrane Ca^2+^ fluxes [88,89]. Thus, they have been intensively studied in mammalian spermatozoa. A first study reported several variants in the *VDAC3* gene that distinguish asthenozoospermic patients from normozoospermic individuals [46,47]. These variants consist of deletions in exons 5 to 8 and missense mutations in exons 6 to 7 (Ile131Leu, Lys171Glu, Asp228Asn), which were not found in control individuals. In addition, an abnormal hypermethylation profile of the *VDAC2* promoter has been suggested to be a pathogenic factor in cases of idiopathic asthenozoospermia [45]. 

Consistent with the pathogenicity of the *VDAC* variants identified in asthenozoospermic men, studies in mouse and bovine models showed that *VDAC2* and *VDAC3* proteins have multiple localization sites: the sperm mitochondrial sheath, the flagellum, and the acrosomal tip (plasma or acrosomal membrane). On the contrary, *Vdac1* appears to be solely expressed in Sertoli cells [90]. *Vdac2* KO mice are not viable due to embryonic mortality [91] and testis- or epididymis-specific KO mice have not been reported. The inhibition of mouse sperm *VDAC2* using DIDS (4,4′-diisothiocyanostilbene-2,2′-disulfonic acid) in vitro led to decreased sperm motility and viability, together with inhibition of tyrosine-phosphorylation during the capacitation process [92], further supporting a role for *VDAC2* in sperm motility and capacitation. In the boar, the role of *VDAC2* during capacitation was also confirmed by a more recent study using the *VDAC2* inhibitors erastin and olesoxime. Erastin treatment was shown to reduce the percentage of non-capacitated spermatozoa, whereas olesoxime prevented the increase in membrane lipid disorder, which constitutes a hallmark of sperm capacitation [93]. *VDAC2* may also have an additional function in the sperm-egg interaction, as it was shown to interact with human recombinant ZP protein in vitro [94]. Interestingly, *VDAC2* constitutes one of the main proteins that were recognized by human anti-sperm antibodies by 2D-electrophoresis, followed by western-blot and mass-spectrometry analysis [94]. Given all these features, *VDAC2* has been classified as a member of the “moonlighting” proteins family, which have different functions in the same cell depending on their specific location. 

*Vdac3* KO mice have been generated and were described by Sampson et al. in 2001. The KO males were shown to be infertile, presenting asthenozoospermia without sperm count defects [95]. Interestingly, those KO mice lack exons 5 to 8, similar to the genomic deletions identified in asthenozoospermic men, suggesting the functional importance of the encoded protein domain. In *Vdac3* KO mice, mature spermatozoa displayed morphological defects such as an anarchic mitochondrial sheath and frequent loss of a single peripheral axonemal doublet. *Vdac3*^−/−^ mice also exhibit abnormal mitochondrial oxidative phosphorylation and a defective mitochondrial shape in the skeletal muscle. It has been suggested that the *VDAC3* protein may also participate in the regulation of the mitochondrial redox state [96]. Mitochondrial dysfunction and altered condensation have also been observed in Drosophila spermatozoa mutated in the *Porin* genes (i.e., *VDAC* orthologs), leading to male infertility [97]. Overall, although there are no strong functional experiments to support the pathogenicity of the *VDAC* variants identified in asthenozoospermic men, the studies in mouse, boar, and Drosophila models provide strong evidence for the function of *VDAC2* and *VDAC3* in the regulation of sperm motility and capacitation. 

### 3.6. SLO3/KCNUI

In mouse, the *SLO3* channel, also called KCNUI, is specifically expressed in the testis and acts as a pH-sensitive channel activated by intracellular alkalization [98]. Invalidation of *Slo3* in the mouse model affects fertility by decreasing progressive motility and acrosomal reaction rates, as well as by impairing membrane hyperpolarization during capacitation. Additionally, the spermatozoa of *Slo3* KO mice are more susceptible to bending and forming hairpin structures under hypotonic stress [99]. Importantly, *SLO3* is considered to mediate the entire pH-dependent K^+^ current in mouse spermatozoa [100]. 

In humans, *hSLO3,* which is specifically expressed in spermatozoa, unlike the K^+^ channel SLO1, is the main candidate channel responsible for the sperm hyperpolarizing K^+^ current [101]. *hSLO3* shares the same sensitivity to alkalinization as the mouse homolog, but also shows voltage- and Ca^2+^-dependent modulation. Geng et al. [101] showed that *hSLO3* is rapidly evolving within the human population and tested the effects of a missense variant allele, which turned out to confer enhanced sensitivity to pH and Ca^2+^. Whether such acquired sensitivity influences fertility potential still needs to be examined. 

Only recently, the screening of sub-fertile patients enrolled in ART confirmed the involvement of *hSLO3* mutations in human asthenoteratozoospermia [48]. Indeed, a Chinese patient carrying the homozygous missense variant p.Ile413Phe showed male infertility due to impaired acrosome formation, mitochondrial dysfunction, altered membrane potential during capacitation and reduced sperm motility. 

### 3.7. Other Ion Channels to Be Explored in Human Asthenozoospermia

*Pharmacological studies on sperm cells*: An important and useful approach to identify ion channels and transporters potentially involved in sperm motility and fertilization potential is to inhibit their functionality in sperm cells. Hence, the use of pharmacological compounds that act as channel inhibitors or activators with a variable affinity spectrum has made it possible to decipher the involvement of ion currents and transport associated with sperm motility and certain hallmarks of capacitation (hyperpolarization, hyperphosphorylation, hyperactivated motility). In this review, the data from the literature related to the use of pharmacological approaches are summarized in Table 2. Among these studies, the potential role of several additional ion channels in human spermatozoa has been highlighted, including that of hHv1, TRPV1, ENaC, NaV, NBC, NCX, CFTR, ANO1, and *hSLO3* (Table 2). 

*Proteomic studies of sperm cells*: Substantial clues have also been provided by comparative proteomic analyses of sperm from infertile patients versus that from normozoospermic individuals. Hence, the first analysis of sperm from asthenozoospermic individuals highlighted the presence of CatSperZ/TEX40, ATP1A4, *VDAC2*, and *TAT1* (*SLC26A8*), together with cation channel sperm-associated protein 1 (CTSR1) in sperm fractions characterized by a high motility profile [122,123,124,125]. Other studies have investigated the role of CLC-3 protein, a Cl^–^/H^+^ exchanger voltage-dependent channel detected in the flagella of human spermatozoa, in the regulation of sperm volume and motility [126]. Hence, spermatozoa from asthenozoospermic men were shown to exhibit an impaired ability to regulate cell volume and decreased motility, such a phenotype is associated with lower CLC3 protein levels than those found in the spermatozoa from normozoospermic individuals [127]. 

*Proteomic studies on seminal vesicles*: Another interesting approach consists of exploring the differential abundance of proteins from seminal plasma vesicles or prostasomes collected from normo- versus asthenozoospermic individuals. Hence, seminal exosomes (including prostasomes and epididymosomes) isolated from samples of normozoospermic individuals were shown to contain higher levels of aquaporin 5 (AQP5), solute carrier family 13 member 2 (SLC13A2), transient receptor potential cation channel subfamily V member 6 (TRPV6), and zinc transporter 4 (SLC30A4) than seminal exosomes collected from patients with severe or moderate asthenozoospermia [128,129]. Similarly, prostasomes from normozoospermic men also showed higher levels of the sulfate transporter SLC26A2 protein than those from asthenozoospermic men [130]. Among other properties, seminal exosomes are considered to be a reservoir of proteins that spermatozoa may acquire after ejaculation, depending on the surrounding environmental conditions, such as the pH (for a review, see [131]). Whether there is fusion between sperm and seminal exosomes is still debated, but protein transfer between these vesicles and spermatozoa has been characterized and shown to influence motility [132,133]. Therefore, sperm fertilization potential could be modulated by ion channels provided by seminal exosomes, in correlation with the decreased contents observed in pathophysiological conditions. 

## 4. Information from Knockout Mouse Models

Current knowledge concerning ion channel function in sperm cells has been mainly acquired via studies in mouse models. Indeed, nearly all pathogenic gene variants that have been described to contribute to asthenozoospermia and male infertility in humans (*CASTSPER*, *SLC26A3*, *SLC26A8*, *SLC9C1*, *VDAC*, *SLO3*) were supported by studies using KO mouse models that show a similar phenotype and pathology. Therefore, it is likely that further substantial improvement of the diagnosis of human asthenozoospermia will rely on information that can be obtained from studies in animal models. In this section, we list all the KO mouse models that, to date, have been reported with a phenotype of asthenozoospermia and for which orthologs in humans could fulfill a similar function. We also describe, when pertinent, the attempts that were made to translate these findings into genetic investigations of human infertility cases.

### 4.1. PMCA4 

PMCA4/ATP2B4 is a plasma membrane calcium ATPase (PMCA), a calmodulin-modulated pump that catalyzes active Ca^2+^ extrusion, thus regulating cellular calcium homeostasis. Its physiological importance for spermatozoa was proven by the phenotype of KO mice in two concomitant studies. Schun et al. reported male infertility due to severe impairment of sperm motility [134], whereas the team of Okunade described hyperactivation defects but normal motility before capacitation [135]. Globally, defective hyperactivated motility and condensed mitochondria were associated with the failure of correct Ca^2+^ clearance and, despite a number of discrepancies between the two studies, PMCA4 appears to be a major regulator of sperm motility and sperm function in the mouse. 

PMCA4 is mainly located on the plasma membrane over the acrosome and the principal piece, the latter reflecting the location of the CATSPER proteins and emphasizing the importance of the fine-tuning of the intracellular sperm calcium concentration for sperm capacitation and fertilization potential [135]. Two splicing variants of PMCA4 (PMCA4a and b) are detected during mouse spermatogenesis, whereas a higher amount of the PMCA4a isoform is observed in murine caudal sperm due to epididymosomal transfer during epididymal transit [136]. Interestingly, PMCA4 was found in oviductosomes secreted by the female genital tract in mice [137], where it may fulfill essential functions for sperm capacitation. The two PMCA4 variants are thought to complementarily participate in a complex with CaM-dependent serine kinase (CASK) and adhesion proteins. 

PMCA4 was also shown to interact with nitric oxide synthases (NOSs) in mouse sperm, mostly under conditions of capacitation, and abnormal levels of reactive oxygen and nitrogen species were recorded in *Pmca4*^–/–^ spermatozoa [138]. The PMCA4-NOSs interaction has been confirmed in human spermatozoa and was also detected in prostasomes [139]. This synergy could participate in sperm regulation at high Ca^2+^ concentrations, preventing oxidative stress, which could impair sperm function and motility. 

Thus far, no gene mutations in PMCA4 have been identified in infertile patients. A study by Stavusis et al. reported no significant association of PMCA4 gene variants in asthenozoospermic patients from Estonian and Latvian populations [140]. 

### 4.2. ATP1A4

ATP1A4 is a catalytic subunit of one of the most important ion pumps of mammalian cells, the Na^+^/K^+^-ATPase. Among the four alpha subunits described, ATP1A4 is the one that shows the most restricted expression pattern, testes being its principal site of expression [141,142]. The ATP1A4 protein is located in the rat sperm flagellum [143], in accordance with the description of its involvement in the regulation of sperm motility [144,145]. ATP1A4 also co-localizes in the rat sperm flagellum with NHE1 and NHE5, two known regulators of intracellular H^+^ content, suggesting a role of ATP1A4 in the regulation of both intracellular pH and motility. Furthermore, ATP1A4 has been reported to be primarily involved in the control of the transport of other ions in rat sperm: the transmembrane Na^+^ gradient, maintenance of the membrane potential, and the regulation of intracellular Ca^2+^ concentration [146]. Not surprisingly, ATP1A4 activity has also been observed to be upregulated during capacitation, in association with increased protein levels at the plasma membrane [147]. This increase during the course of capacitation can be attributed to the mitochondrial translation of ATP1A4 mRNA in spermatozoa, as shown in cows [148]. A mouse model invalidated for ATP1A4 confirmed that ATP1A4 is an essential protein for male fertility, as KO males were completely sterile and their spermatozoa were unable to fertilize oocytes in vitro [149]. In accordance with the observations made using ATP1A4 inhibitors in vitro, spermatozoa from *ATP1A4*^−/−^ mice showed a marked reduction in motility and hyperactivity in association with higher Na^+^ intracellular levels. In humans, ATP1A4 is part of a group of key proteins that were observed to be under-represented in spermatozoa from asthenozoospermic cancer patients relative to sperm cells from healthy fertile men using a proteomic approach [123]. 

### 4.3. TRPV1 and TRPV4

TRPV (transient receptor potential cation) channels were discovered for their sensitivity to increased temperature and proposed to be pain perception mediators [150]. TRPV1 and TRPV4 channels have been detected in spermatozoa from several mammalian species, in which they participate in sperm thermotaxis [151,152]. This response to a thermal gradient is considered to play an important role in sperm progression through the female genital tract, acting as a long-distance guidance system that precedes the chemotaxis known to be exerted at the proximity of the oocyte. As TRPV1 is sensitive to several molecules present in male and female fluids, it may also be involved in regulating the fertilization capacity of sperm by chemotaxis. Although *Trpv1* KO mice have been reported to be fertile [153], a recently obtained zebrafish KO model was described that showed altered fertilization rates without a direct effect on the initial motility ratio or motility decay [154]. Similarly, *Trpv4* gene invalidation in mice did not significantly alter fertility, but interestingly, sperm cells were shown to exhibit delayed initiation of hyperactivation [155]. It is possible that TRPV deficiency may be associated with subtle defects in sperm fertilization potential. Further studies are therefore required to clarify the importance of these channels in sperm function. 

### 4.4. NHA1 and NHA2

NHA1 and NHA2 proteins belong to the SLC9 gene family (SLC9B and SLC9B2 subgroups, respectively) and act as Na^+^/H^+^ antiporter integral membrane proteins. NHA1 was identified in 2010 and proposed to act in the regulation of sperm motility based on functional studies using an anti-NHA1 antibody that decreased sperm motility and the in vitro fertilization rate [156]. In the mouse, *Nha1* invalidation led to decreased fertility (pregnancy rate and litter size). The sperm motility of *Nha1* KO males was significantly reduced, explaining this result [157]. The same phenotype was observed for *Nha2* KO males. The function of *Nha1* and *Nha2* may be partially redundant, as they show high homology. Hence, *Nha2* mRNA levels were elevated in *Nha1* KO mice and vice versa, suggesting a compensation phenomenon. Mice invalidated for both the *Nha1* and *Nha2* genes were completely sterile, with severe asthenozoospermia. A significantly reduced level of cAMP was observed in *Nha1^–/–^*; *Nha2^–/–^* sperm cells, demonstrating severe impairment of the sAC-cAMP signaling pathway [157]. Both channels are expressed in human testes, and the authors demonstrated the presence of NHA1 protein in the human sperm principal piece, indicating that it may be involved in human asthenozoospermia.

### 4.5. SLC9A8 

Within the SLC9 gene family, another Na^+^/H^+^ exchanger has been implicated in male fertility. NHE8 (SLC9A8) is expressed in many murine tissues but in particular in the kidneys, liver, skeletal muscle, and testes. The phenotype of *Slc9a8* KO mice is complex, showing both hormonal and sperm defects. Indeed, *Slc9a8* is strongly expressed in both Leydig and germinal cells. *Slc9a8* deficiency leads to a decrease in LH receptor (LHR) protein levels in Leydig cells, which results in a decrease in testosterone production [158]. In parallel, analysis of the sperm phenotype showed major defects of the head, loss of the acrosome, abnormal distribution of the mitochondria, and severe asthenozoospermia. All of these abnormalities led to sterility in male mice [159]. A conditional KO mouse was generated using the Stra8-iCre mouse line to determine the contribution of NHE8 (SLC9A8) in male germ cells. Germ cell-specific disruption of *Slc9a8* caused a phenotype indistinguishable from that of *Slc9a8*^–/–^ mice, indicating that SLC9A8 protein is mainly required in germ cells for intact acrosome biogenesis and sperm development and motility and, thus, male fertility [159].

### 4.6. CNNM4 

CNNM (former conserved domain protein/cyclin M) constitutes a family of four integral membrane proteins involved in magnesium homeostasis [160,161]. For many years, there has been debate to define whether CNNM proteins are directly involved or simply mediators in the transport of magnesium [162]. Recent studies indicate that CNNMs may modulate the intracellular concentration of magnesium by both means [163,164]. Male mice invalidated for *Cnnm4* were shown to be infertile, with a strong decrease in the pregnancy rate and offspring numbers when mated with wildtype females [165]. The histology of the testes and epididymis, as well as sperm morphology, were completely normal. However, the results of in-vitro fertilization assays were altered in the presence of the zona pellucida and under conditions that support capacitation. Computer-aided sperm analysis highlighted a decrease in sperm motility over time, as well as alteration of the velocity parameters (path and linear and track velocities). Moreover, sperm from *Cnnm4* KO mice were unable to develop hyperactivated motility. Surprisingly, *Cnnm4*-deficient sperm also showed excessive tyrosine phosphorylation during the course of capacitation. This phenotype was very similar to that observed in CATSPER-deficient mice, suggesting an impaired Ca^2+^ influx in *Cnnm4^–/–^* sperm cells during capacitation. An increase in the intracellular Mg^2+^ concentration was also observed in *Cnnm4^−/−^* sperm cells. Importantly, when the spermatozoa were incubated in Mg^2+^-free medium, sperm motility, protein phosphorylation, and hyperactivation were restored to normal levels, indicating an essential role of *CNNM4* in both Mg^2+^ and Ca^2+^ homeostasis [165].

As ***CNNM4*** is also present in the epididymis [166], the authors developed another mouse model specifically invalidated for *Cnnm4* in the germ cells to distinguish between intrinsic sperm and epididymal maturation effects. Sperm isolated from germ cell-specific *Cnnm4*-deficient mice were phenotypically similar to those isolated from total KO mice, confirming that *CNNM4* functionality is crucial within germ cells. Overall, these data show that the regulation of intracellular Mg^2+^ levels is important for sperm motility and hyperactivation, with a possible functional dependence between Mg^2+^ efflux and Ca^2+^ homeostasis. *CNNM* genes have been highly conserved throughout evolution [167], and the presence of *CNNM4* protein was confirmed in human sperm [165], suggesting that mutations in human *CNNM4* may be involved in male infertility. 

Overall, the above studies in mouse models have highlighted several candidate genes for which the mutations may be responsible for human asthenozoospermia. Importantly, a substantial list of other ion channels and transporters have been investigated using KO mouse models, but these studies failed to indicate major functions in the control of sperm motility and male fertility (for review, see [14]). Further studies would be required to define subtle phenotypic and/or compensatory mechanisms to formally exclude their involvement in sperm motility and capacitation events.

## 5. Perspectives in Therapeutics and Male Contraception: Clues from Epididymosomes

Ion channels are very attractive targets for the potential positive or negative regulation of sperm motility. In particular, the search for non-hormonal pharmacological inhibitory compounds for male contraception has recently emerged as an active research field (for reviews, see [168,169]). Among the criteria that should be considered for the selection of protein targets for the development of such pharmacological modulators, maximally restricted tissue distribution is critical to avoid potential systemic negative effects. In addition, the capacity of the pharmacological compounds to reach the sperm cells within the highly immunologically protected testicular environment should also be considered. 

As mentioned above, an important feature of sperm functionality is their post-testicular maturation, which partially occurs during their transit within the epididymis through the interaction of sperm cells with the epididymal epithelium and fluid. A clear illustration of the extent of this process concerns epididymal protein changes and has been well-documented in mouse spermatozoa [170]. Based on a proteomic analysis of spermatozoa from the caput, corpus, and cauda epididymis, Skerget et al. identified 732 proteins that were added to the sperm cells, whereas 1034 proteins were lost from the sperm cells during their descent in the organ. Interestingly, among all proteins concerned by this dynamic maturation process, several ion channels were identified, meaning that dysfunction of the epididymal maturation process could subsequently alter sperm physiology and male fertility. 

The molecular processes involved in the exchange of proteins during epididymal transit are still not finely characterized, but it has been clearly established that small extracellular vesicles secreted from the epididymal epithelium, called epididymosomes, are involved (reviewed by [171]). A proteomic analysis of mouse epididymosomes was recently performed and reinforced the data obtained by sperm proteomic analyses [172] by identifying a list of proteins that are added to the sperm cells during epididymal descent and also detected in the epididymosomes (Table 3). All the proteins reported in Table 3 are located at the sperm plasma membrane, confirming their potential interest as pharmacological and/or immunological targets. Herein, we describe, in more detail, three of these epididymosomal channels delivered as transmembrane proteins to the sperm cells, for which recent studies have attempted to specify compounds that modulate their activity and that are potentially suitable for male contraception.

**Table 3 ijms-23-03926-t003:** Proteins identified by proteomic analyses in both epididymal spermatozoa [170] and mouse epididymosomes [172]. * Proteins are also present in human prostasomes [173,174].

Protein Name	Uniprot Ref	Sub-Cellular Location
ATP2B4 (PMCA4) Plasma membrane calcium-transporting ATPase 4	Q6Q477	Plasma membrane—flagellum membrane
VDAC 1/2/3 Voltage-dependent anion-selective channel protein 1/2/3	Q60932 Q60930 Q60931	Mitochondrion outer membrane—Sperm plasma OR acrosomal membrane—dense outer fibers (VDAC 2/3)
ATP1A1/A4 * Sodium/potassium-transporting ATPase subunit alpha-1/4	Q8VDN2 Q9WV27	Plasma membrane—sarcolemma
CACNA2D1 * Voltage-dependent calcium channel subunit alpha-2/delta-1	O08532	Plasma membrane
ATP1B1/B3 Sodium/potassium-transporting ATPase subunit beta-1	P14094 P97370	Plasma membrane
CLIC1 */3 Chloride intracellular channel protein 1/3	Q9Z1Q5 Q9D7P7	Plasma membrane—cytosol—nucleus
SLC38a5 Sodium-coupled neutral amino acid transporter 5	Q3U1J0	Plasma membrane

### 5.1. PMCA4 

As mentioned above, the critical role of PMCA4 in the regulation of sperm motility was demonstrated through the development of mutant mouse models (Section 3). PMCA4 protein is detected in sperm retrieved from all epididymal segments [170]. Several studies showed that PMCA4 could also be transferred to spermatozoa via epididymosomes in cows [175] and mice [136], as well as via oviductosomes secreted by the oviducts of female mice [137]. The presence of a controlled amount of PMCA4 in sperm cells appears to be a prerequisite for fertilization, making this protein a very interesting candidate in the search for molecular targets to control male fertility [176]. In this regard, the use of PMCA inhibitors to inhibit sperm motility has been the object of a US patent (US20120027815A1) deposited by the same group. Although the protein distribution of PMCA4 is not restricted to the male reproductive system, its activity appears to be highly specialized in spermatozoa. The solving of its three-dimensional structure and the availability of several compounds targeting the ATPase activity contribute to the sustained interest in PMCA4 as an attractive pharmacological target to regulate sperm motility. 

### 5.2. VDAC3 

As previously described, VDACs are voltage-dependent anion channels localized to the mitochondrial outer membrane of spermatozoa and other cell types [90]. Their involvement in sperm motility and physiology has been confirmed by several studies [92,95]. Until now, VDACs were considered to be promising therapeutic targets for cardiovascular and neurodegenerative diseases [177] but a recent review reported the possibility of designing specific *VDAC3* channel blockers to act as male contraceptives, even if the contraceptive potentiality of such channel blockers still needs to be explored [96]. The authors used minocycline (7-dimethylamino-6-demethyl-6-deoxytetracycline), which is an antibiotic, a class of drugs that commonly shows a deleterious impact on spermatogenesis and sperm functionality. They showed impairment of human sperm motility. Various compounds were also tested to block VDAC activity in mouse spermatozoa, including DIDS (4,4′-diisothiocyanostilbene-2,2′-disulfonic acid) and minocycline; both compounds induced a decrease in sperm motility but also presented certain limitation. Hence, although DIDS is known to hamper proper channel function and the organization of VDACs, it is also known to interfere with other ion transport complexes. The possibility of using antibiotics with weaker effects on spermatogenesis, such as tetracyclines, which have been proven to interact with *VDAC3*, may be an interesting approach for the development of an efficient male contraceptive strategy. For such strategies, it will be fundamental to better characterize the function of *VDAC3* in tissues other than the testes to prevent undesirable side effects.

### 5.3. ATP1A4

As described in Section 3, ATP1A4 appears to be a very interesting candidate for male contraception. These aspects have been recently reviewed [178], putting the emphasis on the possibility to pharmacologically block this sperm-specific isoform. Ouabain has the ability to bind to Na^+^/K^+^-ATPases but is not specific to ATP1A4, which led the authors to test more selective compounds of the ouabain family, but without success. Synthetic molecules were also tested and one compound named “compound **25**” showed an efficient in vitro inhibitory effect on rat epididymal sperm motility at 10 nM and higher concentrations. Interestingly, this compound also efficiently inhibited sperm hyperactivated motility by 75%. “Compound **25**” had additional effects on ion-dependent parameters; it caused sperm plasma membrane depolarization, reduced cytosolic pH by 15%, and increased the intracellular Ca^2+^ concentration by 40%. Aside from its in vitro effects, daily oral gavage at a dose of 5 mg/kg in rats led to the inhibition of epididymal sperm motility after three days of treatment. Importantly, this compound provoked an 80% decrease in the ability of sperm to fertilize oocytes in vitro. The authors mentioned that in vivo mating experiments were under assessment, which should provide important data to prove the relevance of “compound **25**” for male contraception.

Importantly, the involvement of other ion channels not described to be present on either spermatozoa or post-testicular membrane vesicles have been reviewed lately and also constitute good candidates for male contraception [14,168].

## 6. Concluding Remarks

Over the last decade, the emergence of high-throughput sequencing technologies has allowed tremendous advances in the field of reproductive biology and the identification of several genetic causes of male infertility. However, although asthenozoospermia is the predominant sperm pathology found in infertile men, as documented here, the genetic causes for this condition remain very limited and only account for a few patients. Importantly, most often, these cases correspond to severe asthenozoospermia, while the predominant context of asthenozoospermia is found moderated and often in association with sperm count and morphological defects. Further studies in the field are therefore required to maximize the genetic diagnosis that can be offered to the patients. In addition, the improvements in genetic diagnosis are still not accompanied by a treatment. The only solution currently proposed is still intracytoplasmic sperm injection (ICSI), which involves substantial medical procedures for the partners of infertile men (hormonal injections to induce superovulation and ovary puncture to collect the oocytes, followed by embryo transfer to the uterus after ICSI) and in vitro culture of the gametes. Therefore, the identification of potent strategies to stimulate sperm motility and fertilization potential, in vitro or in vivo, could represent an alternative to ICSI by the direct treatment of infertile men. The literature review we performed here strongly confirms the existence of several ion transmembrane channels that could be good targets for the development of such novel therapeutical strategies in the near future. Importantly, targeting these same proteins using specific inhibitory compounds could also be a key entry point for male contraception.

## Figures and Tables

**Figure 1 ijms-23-03926-f001:**
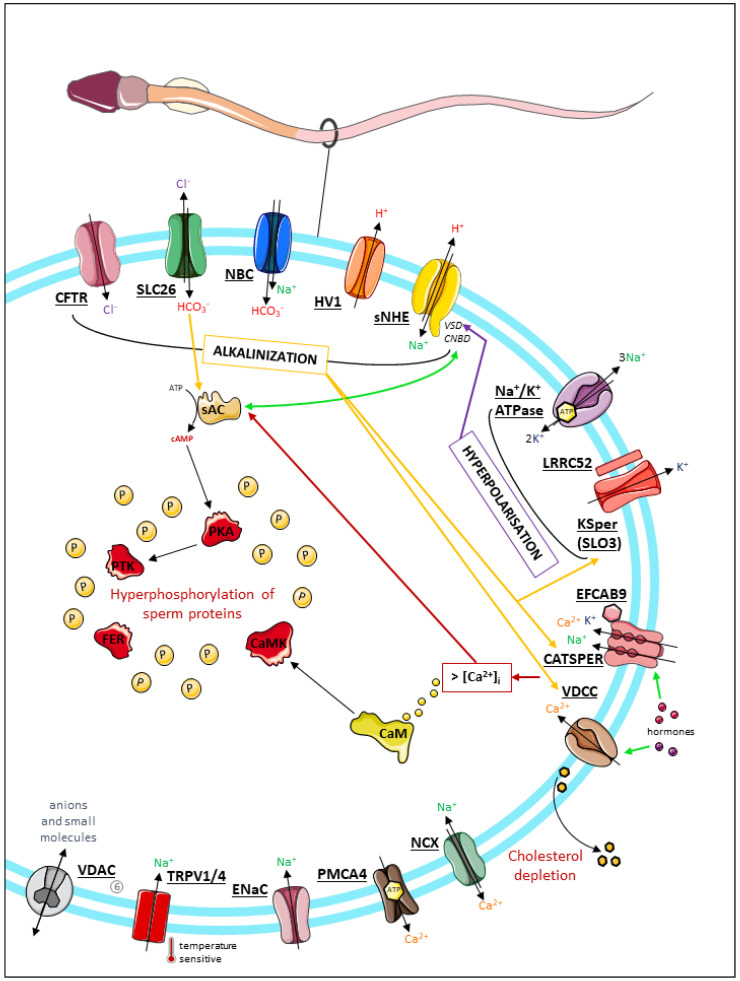
Scheme summarizing the main molecular, biochemical, and electrophysiological events associated with sperm capacitation and the main ion channels/transporters involved in the process. The first subset of ion channels (SLC26 family interacting with CFTR, NBC, Hv1, and sNHE) are responsible for intracellular alkalinization of the sperm by internalizing bicarbonate or extruding protons. The subsequent pH increase is responsible for sAC activation, which induces an increase in intracellular cAMP. cAMP is a key element of sperm signaling during capacitation, in particular, by mediating PKA-dependent phosphorylation events and also by promoting the activity of other ion channels, such as CatSper, VDCC, and KSper (yellow arrows). CatSper and VDCC mediate Ca^2+^ entry into spermatozoa and are sensitive to the stimulation of progesterone and other hormones present in the female genital tract. The increase in intracellular Ca^2+^ concentration also participates in the activation of sAC (red arrow). In addition, membrane hyperpolarization by cation extrusion through KSper and Na^+^/K^+^ ATPase also influences voltage-sensitive actors, such as sNHE (violet arrow). It also temporally regulates the activity of other channels, as it inhibits CatSper and Hv1 (Vyklicka and Lishko, 2020). Of note, Hv1 is absent from murine spermatozoa; this voltage-gated channel is specific to human sperm cells, in which it is restricted to the principal piece of the flagellum within two bilateral longitudinal lines asymmetrically interacting with the four CatSper nanodomains [26]. CFTR, cystic fibrosis transmembrane conductance regulator; SLC26, solute carriers family 26; NBC, Na^+^/HCO_3_^−^ cotransporter; Hv1, voltage-gated H^+^ channel; *sNHE*, sperm-specific Na^+^/H^+^ exchanger; LRRC52, leucine-rich repeat-containing protein 52; KSper, K^+^ sperm channel; EFCAB9, EF-hand Ca^2+^-binding domain-containing protein 9; CatSper, cation sperm channel; VDCC, voltage-dependent Ca^2+^ channel; NCX, Na^+^/Ca^2+^ exchanger; PMCA4, plasma membrane Ca^2+^ ATPase4; ENaC, epithelial Na^+^ channel; TRPV4, transient receptor potential cation channel subfamily V member 4; VDAC, voltage-dependent anion channel. This picture contains graphical elements obtained from smart.servier.com (accessed on 20 March 2022).

**Table 1 ijms-23-03926-t001:** Pathogenic variants in genes encoding ion channels or transporters and associated with asthenozoospermia and male infertility in humans, AST, asthenozoospermia; OAT, oligo-astheno-teratozoospermia; AZOO, azoospermia; HET: heterozygous; HOM: Homozygous; ICSI, intracytoplasmic sperm injection; IVF, In vitro fertilization; N/A: not applicable; PGD, preimplantation genetic diagnosis.

Ion Channel	Mutation(s)		Patients	Phenotype	ART Outcome	Reference
*CATSPER1*	c.539–540insT (p.Lys180LysfsX8) c.948–949insATGGC (p.Asp317MetfsX18)	HOM	3 patients from 2 consanguineous Iranian families	OAT Low semen volume	-	Avenarius et al., 2009 [34]
*CATSPER2*	~70-kb deletion in chr 15q15 (last 2 exons of *CATSPER2*)	HOM	3 brothers from a French family	Congenital dyserythropoietic anemia type I (CDAI) Astheno-teratozoospermia (91% sperm with short, coiled flagella)	-	Avidan et al., 2003 [35]
			1 patient	Severe asthenoteratozoospermia Disruption of progesterone-sensitive Ca^2+^ current Absence of Catsperβ	-	Avidan et al., 2003 [35]; Smith et al., 2013 [36]
	~100-kb deletion in chr 15q15.3 (integral loss of *CATSPER2*)	HOM	7 patients from 3 Iranian families (one of which is consanguineous)	Deafness infertility syndrome (DIS) Astheno-teratozoospermia (thin heads, macrocephaly, irregular acrosome, short or coiled flagella)	-	Zhang et al., 2007 [37]
				Quadrilateral CatSper domains, sperm rolling (rotation around the longitudinal axis), and rheotaxis (navigation in fluid flow) are not affected	-	Schiffer et al., 2020 [38]
	CNV (one copy lost in the region of 43,894,500 to 43,950,000 on chr 15q15.3, containing the entire *CATSPER2)* deletion of *CATSPER2*	HET	1 Chinese patient	Normal semen parameters Impaired sperm hyperactivation, no response to progesterone	Failure of IVF cycle (6 oocytes) ICSI positive outcome with singleton pregnancy	Luo et al., 2019 [39]
*CATSPERε*	In-frame 6-bp deletion in exon 18 (c.2393_2398delCTATGG, p.Met799_Ala800del)	HOM	1 European patient	Normal motility and concentration Loss of CATSPER function	Failure of IVF	Brown et al., 2018 [40]
*SLC26A3*	V317del	HOM (7) HET (1)	8 Finnish patients	Congenital chloride diarrhea (CLD) with male subfertility (OAT) Spermatoceles	-	Höglund et al., 2006 [41]
	c.2062 G > C (p.Asp688His) (9) c.949_951delGTG (p.Val318del) (3)	HET	12 Finnish patients	11/12: severe or moderate oligo-astheno-zoospermia 1/12: azoospzermia (patient also carrying a CFTR 5T heterozygous allele)	-	Wedenoja et al., 2017 [42]
*SLC26A8 (TAT1)*	Patient 1: c.260G > A (p.Arg87Gln) (1) Patient 2: c.2434G > A (p.Glu812Lys) (1) Patient 3: c.2860C > T (p.Arg954Cys) (1).	HET	3 patients Patient 1: Caucasian Patient 2: North African Patient 3: Caucasian	Astheno-teratozoospermia (Patient 3 has OAT) Reduced amount of SLC26A8-mutated protein and associated CFTR channel	Patient 1: ART attempt, natural pregnancy after 2.5 years without proof of paternity Patient 2: ART attempt, natural pregnancy with a new partner Patient 3: ART attempt, 4 ICSI failures	Dirami et al., 2013 [43]
*SLC9C1 (sNHE)*	c.2748 + 2T > C (p.Glu884_Lys916del)	HOM	1 African patient	Asthenozoospermia	ICSI failure	Cavarocchi et al., 2021 [44]
*VDAC2*	CpGs hypermethylation in the promoter		25 Chinese patients	Idiopathic asthenozoospermia	-	Xu et al., 2016 [45]
*VDAC3*	Missense variants in exon 6 (Ile131Leu, Lys171Glu)		-	Asthenozoospermia	-	Asmarinah et al., 2005 [46]
	3 patients: deletion in exon 5 1 patient: deletion in exon 7 1 patient: deletions in exons 5 and 7 1 patient: missense mutation (Asp228Asn) in exon 7 1 patient: deletions in exons 5, 7, and 8		7 Chinese patients	Asthenozoospermia	-	Asmarinah et al., 2011 [47]
*SLO3*	Missense variant c.1237A > T, p.Ile413Phe	HOM	1 Chinese patient	Asthenoteratozoospermia	ICSI positive outcome with singleton pregnancy	Lv et al., 2022 [48]
*PKD1-2*	Several *PKD1* or *PKD2* mutations	N/A	46 Chinese patients	Autosomal dominant polycystic kidney disease (ADPKD) 37/46 patients with abnormal semen parameters (AST, OAT, AZOO)	2 natural pregnancies 2 ICSI 31 ICSI + PGD	He et al., 2018 [49]

**Table 2 ijms-23-03926-t002:** Studies and characterization of ion channels by pharmacological inhibition or activation on human spermatozoa. AR, acrosomal reaction; AC, adenylate cyclase; sAC, soluble adenylate cyclase.

Ion Channel	Current	Inhibitor	Effect	Reference
hHv1	H^+^	hanatoxin-containing venom (*Grammastola rosea*)	No significant change in sperm hyperactivation A combination of venom and progesterone caused decreased full 360° rotation of the sperm flagella	Lishko et al., 2010 [102] Miller et al., 2018 [26]
		Corza6 (de novo peptide inhibitor)	AR inhibited by ∼70% No effect on sperm viability, sperm motility, or tyrosine phosphorylation pattern	Zhao et al., 2018 [103]
		Pantoprazole (proton-pump inhibitor)	Decreased sperm progressive motility and capacitation-induced sperm hyperactivation (hyperpolarization and protein phosphorylation)	Escoffier et al., 2020 [104]
CatSper1	Cations (Ca^2+^)	anti-CatSper1 IgG antibody (H-300)	Reduced sperm progressive motility after 1, 2, and 4 h of incubation Reduced sperm hyperactivated motility after 5 h of incubation	Li et al., 2009 [105]
CatSper		Ketamine	No effect on sperm viability, capacitation, or spontaneous AR Reduced intracellular calcium concentration	He et al., 2016 [106]
		NNC	Reduced sperm viability, motility, and velocity Inhibition of progesterone induced AR	Ghanbari et al., 2018 [107]
		Trequinsin hydrochloride	>hyperactivation and penetration into viscous medium <intracellular cGMP	McBrinn et al., 2019 [108]
CatSper and Hv1	Cations	NNC, ZnCl2, NNC + Zn	Reduced sperm viability, motility, and curvilinear velocity in all groups containing NNC, zinc, and NNC + zinc. The progesterone–induced acrosome reaction was abolished by each of these inhibitors. The combinatory effect of NNC plus zinc on motility and progesterone–induced acrosome reaction was no stronger than NNC by itself.	Keshtgar et al., 2018 [109]
CatSper and *hSLO3*	Cations	RU1968	Reduced progesterone-induced sperm hypermotility Minor inhibitory effect on *hSLO3* rather than CatSper	Rennhack et al., 2018 [110]
TRPV1	Cations	Capsazepine	Inhibition of progesterone-promoted sperm-oocyte fusion Reduced progesterone-induced AR rate, reduced spontaneous AR rate No effect on sperm motility	Francavilla et al., 2009 [111]
ENaC	Na^+^	EIPA	Improved sperm motility in healthy donors and asthenozoospermic patients	Kong et al., 2009 [112]
NaV	Na^+^	Lidocaine	Induction of hyperactivated motility	Candenas et al., 2018 [113]
NHE1/SLC9A1	Na^+^/H^+^	EIPA	No effect on AR	Garcia and Meizel, 1999 [114]
NBC	Na^+^/HCO_3_^−^	S0859	Lower PKA activity	Puga Molina et al., 2018 [115]
NCX	Na^+^/Ca^2+^	bepridil, DCB, KB-R7943	Impaired sperm motility	Krasznai et al., 2006 [116]
CFTR	Anions (Cl^−^)	CFTRinh-172	Inhibition of progesterone-induced sperm capacitation, cAMP synthesis, hyperactivated motility, and rhuZP3a-induced AR	Li et al., 2010 [117]
Cl^−^ channels	Cl^−^	Adjudin (Cl^−^ channels blocker)	Reduced sperm hyperactivation but no effect on sperm motility Blockage of rhuZP3b- and progesterone-induced AR in a dose-dependent manner Inhibition of forskolin-activated transmembrane AC and sAC activity Impaired serine and threonine sperm protein phosphorylation Prevention of sperm penetration of zona-free hamster eggs	Li et al., 2013 [118]
TMEM16A/ANO1	Cl^−^	NFA, DIDS, TMEM16A_inh_	Reduced rhZP3-induced AR	Orta et al., 2012 [119]
K channels	K^+^	Quinine	Increased sperm volume, reduced sperm kinematics and mucus penetration K-ionophores valinomycin and gramicidin counteract 4-aminopyridine but not TEA (K-blockers) Can mimic quinine	Yeung and Cooper, 2001 [120]
*hSLO3*	K^+^	Progesterone, Ba2þ and Quinidine (+++) Penitrem A and Charybdotoxin (+) Iberiotoxin and Slotoxin (~)	Pharmacological comparison of the CAH and *hSlo3* profiles indicates that in addition to *hSlo3*, other K^+^ channels (possibly Slo1) may participate in CAH	Sanchez-Carranza et al., 2015 [121]
NaV	Na^+^	Veratridine	>sperm progressive motility without producing hyperactivation >protein tyrosine phosphorylation Blockage of progesterone-induced AR Membrane depolarisation	Candenas et al., 2018 [113]

## Data Availability

Not applicable.

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
