# Peer review of "Sperm Ion Transporters and Channels in Human Asthenozoospermia: Genetic Etiology, Lessons from Animal Models, and Clinical Perspectives"

_ijms, 2022, doi:10.3390/ijms23073926_

Round 1
Reviewer 1 Report
The authors present a complete overview of genetic anomalies associated with defective sperm physiology.
The statement that 80% of male infertility involves abnormal sperm motility is misleading and not consistent with the rare cases where a specific cause for male infertility can be explained by a specific genetic defect. In toto, there are only case reports where abnormal sperm motility is explained in human infertile males. The authors should be more retrospect in attributing sperm ion channel defects as a common cause of male infertility. Most men with abnormal motility have a global defect in sperm production and quality.
Author Response
As recommended by Reviewer 1, we have added a sentence in the conclusion in order to clarify to the readers that although asthenozoospermia is the predominant sperm pathology found in infertile men, the genetic causes we described in the manuscript only account for a minority of the cases. Most often these are cases with severe asthenozoospermia while indeed the predominant context of asthenozoospermia is found moderated and in association with sperm count and morphological defects as stated by the reviewer. Overall, this statement also stresses that still a lot of work is required to define the genetic causes of asthenozoospermia.
Reviewer 2 Report
The article by Cavarocchi et al is a review of genes/proteins functioning as ion channels and transporters in relation to male infertility. The authors, experts of the field, list the genes for whom mutations have been identified in human cases of asthenospermia and complete this list with genes demonstrated thanks to animal models to participate to the process. The review is exhaustive and extended by interesting perspectives on the use of these proteins as therapeutic targets as well as avenues for contraceptive strategies. It will be useful for clinicians interested in the topic, geneticists and even to a broader public.
I have no real critics about the paper.
I would suggest to add a recent reference showing a mutation in the SLO3 gene in asthenozoospermia that completes this catalog:
Lv et al, 2022, Reprod Biol Endocrinol
Minor suggestions:
- Table 1 could be improved by adding a column describing the status of the patient, heterozygous or homozygous, for each mutation. A discussion could be added in the text to comment on heterozygosity.
- Page 10, line 10 : sheath instead of sheet
- Table 1 : Ref #65 is not correct, it is rather the other Hihnala et al 2006, in Mol Hum Reprod
- Table 2 : Fill the line of RU1968 and move the last line (trequinsin) together with other CatSper inhibitors
Author Response
=> We thank the reviewer for his comments. We have corrected the manuscript following all the recommendations:
I would suggest to add a recent reference showing a mutation in the SLO3 gene in asthenozoospermia that completes this catalog: Lv et al, 2022, Reprod Biol Endocrinol
=> The description of SLO3 mutation recently identified was added (Lv et al, 2022, Reprod Biol Endocrinology)
Minor suggestions:
- Table 1 could be improved by adding a column describing the status of the patient, heterozygous or homozygous, for each mutation. A discussion could be added in the text to comment on heterozygosity.
=> We have added the status of the mutation and we now discuss the heterozygosity of SLC26A3 and SLC26A8 variants. The following comment was added: “Of note, most the above variants identified in SLC26A8 and SLC26A3 were found at the heterozygous state, in contrast to mutations identified at the homozygous state in other SLC26 members, which follow an autosomal recessive mode of inheritance. This difference could be related to the haploid state of the germ cells; indeed, although permanent connection through intercellular bridges allows sharing between the spermatogenic cells, it could be uncompleted and could contribute to a heterogeneous sperm cell population in a given individual. Consistent with this, most individuals carrying heterozygous variants in SLC26A8 and SLC26A3 also presented with mild asthenozoospermia phenotype. In addition, as the heterozygous variants identified in SLC26A8 and SLC26A3 are missenses, they could also behave as dominant negative effect and therefore be pathogenic at the heterozygous state.”
- Page 10, line 10 : sheath instead of sheet
=> This was corrected
- Table 1 : Ref #65 is not correct, it is rather the other Hihnala et al 2006, in Mol Hum Reprod
=> Here the reference is correct as this is the original publication in which the infertility of several CLD patients was investigated and found to be impaired with in most cases oligo-astheno-teratozoospermia together with high chloride and low pH in seminal plasma.
- Table 2 : Fill the line of RU1968 and move the last line (trequinsin) together with other CatSper inhibitors
=> This was corrected